# Plasma dye coating as straightforward and widely applicable procedure for dye immobilization on polymeric materials

Lieselot De Smet[1], Gertjan Vancoillie[1], Peter Minshall [1], Kathleen Lava[1], Iline Steyaert[2], Ella Schoolaert [2], Elke Van De Walle[3], Peter Dubruel[3], Karen De Clerck[2] & Richard Hoogenboom [1]

Here, we introduce a novel concept for the fabrication of colored materials with significantly reduced dye leaching through covalent immobilization of the desired dye using plasma-generated surface radicals. This plasma dye coating (PDC) procedure immobilizes a pre-adsorbed layer of a dye functionalized with a radical sensitive group on the surface through radical addition caused by a short plasma treatment. The non-specific nature of the plasma-generated surface radicals allows for a wide variety of dyes including azobenzenes and sulfonphthaleins, functionalized with radical sensitive groups to avoid significant dye degradation, to be combined with various materials including PP, PE, PA6, cellulose, and PTFE. The wide applicability, low consumption of dye, relatively short procedure time, and the possibility of continuous PDC using an atmospheric plasma reactor make this procedure economically interesting for various applications ranging from simple coloring of a material to the fabrication of chromic sensor fabrics as demonstrated by preparing a range of halo-chromic materials.

[1] Supramolecular Chemistry group, Centre of Macromolecular Chemistry (CMaC), Department of Organic and Macromolecular Chemistry, Ghent University, Krijgslaan 281 S4, 9000 Ghent, Belgium. [2] Centre for Textile Science and Engineering, Department of Materials, Textiles and Chemical Engineering, Faculty of Engineering and Architecture, Ghent University, Technologiepark 907, 9052 Ghent, Belgium. [3] Polymer Chemistry & Biomaterials Research Group, Department of Organic and Macromolecular Chemistry, Ghent University, Krijgslaan 281 S4Bis, 9000 Ghent, Belgium. These authors contributed equally: Lieselot De Smet, Gertjan Vancoillie. Correspondence and requests for materials should be addressed to R.H. (email: richard.hoogenboom@ugent.be)

The process of coloring materials has been around for many decades and is used for a wide variety of applications with the simplest examples including the dyeing of fabric in order to apply different colors. Recently, colored materials are gaining interest for their applicability in the field of optical sensors with these 'smart' sensory materials responding to a small environmental change with a clear and immediate color change[1–6]. Colorimetric sensors are most commonly fabricated by attaching analyte-sensitive dye molecules to a carrier material or surface, resulting in a change in optical properties under the influence of numerous stimuli including ions, gasses, and a wide range of volatile organic compounds (VOCs)[7–11]. The straightforward use and unambiguous output signal of these sensors allow for their implementation in quick sample analysis but also continuous monitoring for a specific analyte. The visualization of the pH as a fast and easy-to-interpret colometric signal can prove useful for various applications in the biomedical field, agriculture, safety, and technical textiles since pH plays a major role in many procedures including wound healing, crop growth, and filter performance[12,13]. A specific example is the implementation of such a material in a continuous shelf-life sensor for fish or meat. The spoilage of fish or meat is usually accompanied by an increase of microorganisms that produce various volatile amines like trimethylamine and ammonia, collectively called total volatile basic nitrogens (TVB-N). By incorporating the pH-responsive Bromocresol Green into the sensor, the increase of TVB-N in the headspace of the packaging can visually be monitored by a color change from yellow to green and act as a warning sign for potential buyers[14].

The two most common methods for the fabrication of colored materials is to introduce the dye molecule into the feed-mixture of the fabrication process, so-called dye-doping, or to apply the dye after fabrication as a dyeing step[12,15]. Both methods are fairly simple and compatible with a wide variety of dyes or pigments, polymer materials, and fabrication processes including hot melt extrusion, injection molding, and electrospinning[16–22]. A major drawback of post-fabrication dyeing is that after this wet process the dye is only loosely attached to the material surface by means of physical interactions. This can result in significant leaching of the dye when the material comes into contact with liquids or other materials resulting in decreased sensor sensitivity or output signal and possible toxicological response in biomedical applications[22]. The most common approach to reduce the dye leaching and to increase the long-term sensitivity and stability of the sensor is the addition of a polymeric fixating agent like Perfixan®. This polycation is used to immobilize anionically charged dyes through electrostatic interactions[13,22,23]. The latter however does not completely suppress dye leaching and provides only solutions for very specific applications or material/dye combinations.

The only way to fully suppress dye leaching from colored materials is to immobilize the dye on the polymer material through covalent linkage which can again be performed both pre- and post-fabrication[23–27]. In the pre-fabrication pathway, the dye immobilization is achieved through the functionalization of a precursor polymer with the desired dye after which the dye-functionalized polymer is introduced into the feed-mixture of the fabrication process. This allows for the production of homogenously colored materials in which the dye is strongly held in the bulk of the material through polymer–polymer entanglements resulting in significantly reduced dye leaching. Our group has successfully employed this approach to fabricate halochromic non-woven materials using blend-electrospinning from polyamide-6 (PA6)[28], poly-ε-caprolactone (PCL)[29], and silica-based fibers[30]. The main disadvantage to this approach is the large scale and sometimes difficult modification of the dye especially for the more interesting and complex dye molecules.

The material-specific fabrication conditions that require optimization for each polymer blend and dye combination also make this approach time-consuming and costly in both development and production, restricting it to high-end applications like biomedically compatible sensory materials. Comparatively, a post-fabrication modification pathway for the covalent immobilization of dye molecules to the surface of the material has far more economic potential due to the possibility of using existing, optimized fabrication methods with reduced dye consumption as only the surface of the material is modified. A common strategy to achieve this covalent linkage is through the use of specific surface functionalities including amines and alcohols allowing the immobilization through respectively amide or ester formation which is also employed for other molecules including boronic acid groups[31], polymeric initiators[32–35], or chain transfer agents[36]. This approach is however restricted to reactive surfaces, such as silica and cellulose, or require the incorporation of precursor compounds during fabrication to introduce specific functionalities. Residual unreacted groups can also cause compatibility issues during application, making it not a generally applicable approach especially not for common, inert plastics like polyethylene (PE), polypropylene (PP), or polytetrafluoroethylene (PTFE).

An alternative method for covalent immobilization that bypasses the necessity of a specific combination of reactive groups is based on the use of surface radicals on the material, thereby expanding the range of applicable materials. After the generation of radicals on the surface through proton abstraction or homolytic cleavage of the weakest bond, a new covalent bond between surface and additive can be formed through radical recombination or radical addition reactions. An example of this approach was published recently by Lee et al.[37] using high-energy UV-irradiation with a specific wavelength (254 nm) in order to homolytically cleave the labile carbonyl double bond in PA6 and subsequently immobilize styrene and allyl-functionalized catalysts. The applicability of this method is however limited by the expensive irradiation equipment, high material specificity, and incompatibility with light absorbing dyes that rapidly degrade under the used conditions. An alternative, cheaper and more generally applicable approach for the generation of surface radicals can be found in the use of plasma surface modification (PSM) processes. Plasma consists of a gaseous mixture of highly energetic ionized atoms, radicals, electrons, and neutral molecules, which can interact with the surface of various substrates. Commonly used gases include oxygen, argon, and helium, which can be energized using for example extreme heat or by the application of an electrical field[38]. While thermal plasmas are often used in molecular characterization methods like mass spectrometry and optical emission spectroscopy, the non-thermal or cold plasmas are more applicable for the treatment of material surfaces especially of a polymeric substrate as the extreme heat would degrade the material[39]. In a recent review by Khelifa et al., cold plasma techniques are divided into two main categories: non-depositing and depositing techniques[40]. Non-depositing techniques, commonly known as plasma etching, employs plasma to clean material surfaces by removal of a few layers of atoms and possible surface contaminants[38,41,42]. Depositing techniques or plasma-enhanced chemical vapor deposition (PECVD) are used to introduce novel functionalities on the surface of the material by injecting various organic molecules into the plasma[40,43]. This allows for control over surface properties like hydrophilicity, biocompatibility, and adhesiveness but also the surface chemistry by introducing reactive functionalities using simple molecules like amines (NH$_3$-plasma, 1,3-diaminopropane)[42,44], alcohols and epoxides (O$_2$-plasma)[41,45], and fluorine groups (C$_2$F$_6$-plasma)[46]. These reactive surfaces can subsequently be used for

immobilization of complex biomolecules such as biotin[47] but also for surface-initiated polymerization[48–50].

Plasma depositing techniques are however not limited to simple gaseous molecules but can also be used to introduce much more complex organic molecules in a one-step approach during the plasma treatment. This approach was reported as early as 1992 by Hoffman et al.[51,52] as Crosslinking by Activated Species of INert Gases (CASING) in order to immobilize decylamine[51] or sodium dodecylsulfate[52] on, respectively, PP or PET. In this method, the additive is absorbed on the surface of the material after which plasma treatment covalently couples the molecule to the substrate through radical interactions with plasma-generated surface radicals through hydrogen abstraction or homolytic cleavage of the weakest bond[51–53]. This method was also successful for the immobilization of both saturated and unsaturated surfactants including sodium 10-undecenoate and sodium dodecanoate on PE, PP, and polycisbutadiene[54–56], but also for more complex additives including poly(N-vinyl-2-pyrrolidone) on PP membranes[57] and various functionalized alkylated-poly (ethylene glycol) (PEG) derivatives[58].

A final deposition technique is plasma polymerization where a thin film is directly formed onto the substrate during exposure to plasma[59,60]. In this process, gaseous or pre-absorbed simple monomers are fragmented via inelastic collisions with the energetic plasma, which are consecutively recombined and bounded into a thin film formed on the material surface. Major drawback of this technique is that the composition as well as the incorporated functionalities of the highly cross-linked film cannot be predicted due to the high fragmentation rate, leading to drastic alternation of the surface properties. Recently, Barranco and coworkers[61] reported a specific PECVD technique to produce highly luminescent polymeric films by sublimating and activating dye molecules downstream the plasma reactor lowering the fragmentation rate. Major drawbacks of this procedure are the complicated geometry of the plasma chamber and the drastic changes in surface properties resulting from the formation of a dye film[61,62].

In this manuscript, we report the development and optimization of the plasma dye coating (PDC) procedure, a simple method that employs the use of a plasma depositing technique to covalently immobilize pre-absorbed dyes that are functionalized with a radical sensitive group. It is demonstrated that the radical sensitive group, in the form of a double bond that is incorporated into the dye structure, is required and acts as an antenna for the reactive species generated by plasma leaving the majority of the dyes intact during the PDC process. This allows the fabrication of colored materials with significantly reduced dye leaching starting from a wide variety of base materials due to the non-specific nature of the plasma surface-radical generation, whereby the presence of the radical sensitive group is crucial to suppress dye degradation. Subsequently, we have employed the optimized procedure to highlight the wide applicability of the PDC procedure by modifying six different materials including PA6, cellulose, PP, Teflon, high-density and low-density PE with five different dyes including sulfonphthaleins, azobenzenes, and dimethylamino-naphthilimides using the same general surface functionalization protocol.

## Results

**Optimization of the PDC procedure.** The effect of plasma treatment on a polymeric material results from two competitive reactions namely surface degradation and additive immobilization, which in turn control whether molecules are being incorporated or removed from the surface, i.e., deposition or etching. While the latter is favored in dense, mostly ionic plasmas with

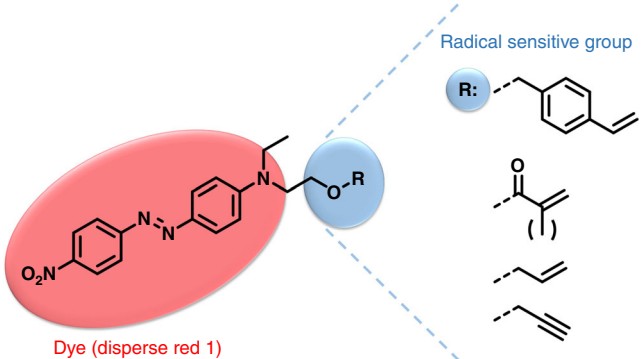

**Fig. 1** Structure of the dye disperse red 1 (DR1) with the different radical sensitive groups. (Red) The chemical structure of DR1. (Blue) The radical sensitive groups necessary for the covalent immobilization, from top to bottom: styryl, (meth)acrylate, allyl, and propargyl moieties

long treatment times, the additive immobilization or cross-linking of molecules on the material surface is favored during short plasma treatments of a few minutes with inert gas plasma containing mostly radical species. Besides the plasma gas and/or possible non-volatile additives, we identified the plasma density and plasma treatment time as the two main instrumental parameters of the plasma equipment[41,45,63,64]. When using complex molecules like dyes with their highly delocalized structures resulting in unique electronic spectra, the degradation of the dye due to the highly energetic plasma becomes an additional reaction competing with the successful irreversible coloring of the material. In order to keep the structure of the dye intact, additional groups were incorporated that presumably would preferentially react with the formed surface radicals, leading to covalent immobilization of the dye rather than its degradation. These radical sensitive groups are also commonly used in various monomers for radical polymerization and include styryl, (meth) acrylate, allyl, and propargyl moieties (Fig. 1). For the optimization of the PDC procedure, we selected disperse red 1 (DR1) because of its commercial availability, strong red color and easy modification using the isolated aliphatic primary alcohol group. This functional group allows for the incorporation of various moieties through ester- and etherification reactions while limiting the effect on the electronic properties (Supplementary Note 3).

The so-called PDC procedure consists of three simple steps (Fig. 2): (1) the material is dipped in a solution of the desired polymerizable dye in a volatile solvent like ethanol or THF to allow a homogenous adsorption of the dye on the material surface. The relative short dipping time in combination with a non-solvent limits dye diffusion into the polymer material. (2) The material is treated for a short period of time with a non-thermal plasma to allow the covalent immobilization of the dye through radical addition with the generated radicals on the surface of the material, on the radical sensitive group or both. (3) Any residual, unreacted dye or possible side compounds are removed using soxhlet extraction. Initially, four instrumental parameters were identified that could play a vital role in the successful immobilization of the dye on the material: (1) the dipping time, (2) the dye concentration in the dipping solution, (3) the plasma treatment time, and (4) the identity of the radical sensitive group. We optimized each of these parameters sequentially using the combination of a DR1 derivative and a non-woven electrospun PA6 substrate. The dye was chosen because of its easy, large scale synthesis and wide variety of possible polymerizable groups while the PA6 material allows for the quantification of the eventual dye loading through solution UV-VIS spectroscopy in FA/AA (1/1) in which the modified PA6

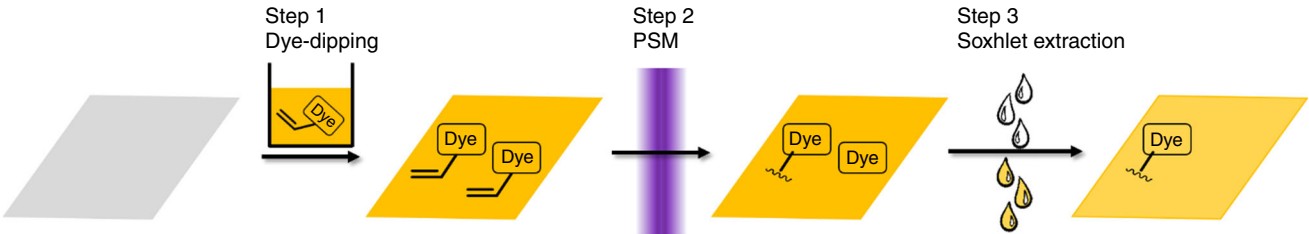

**Fig. 2** Schematic representation of the plasma dye coating procedure

**Fig. 3** Optimization of the plasma treatment time. **a** Dye loading on PA6 surface after PDC treatment measured by UV-VIS spectroscopy in 1/1 FA/AA in function of plasma treatment time on a single side of a 1-cm$^2$ piece of material in triplicate. **b** Pictures of wetted samples of PDC treated PA6 with DR1-A compared to untreated PA6 after soxhlet extraction for different plasma treatment times

**Fig. 4** Influence and importance of the radical sensitive moiety. **a** Dye loading on PA6 surface after PDC treatment measured by UV-VIS spectroscopy in 1/1 FA/AA with varying polymerizable dyes and additives. All tests were performed on a 1-cm$^2$ piece of material in triplicate. **b** Proof-of-principle of the necessity of PDC procedure comparing the optimized PDC procedure and an altered procedure where key-components are removed including no polymerizable group and no plasma treatment. All pictures are taken adjacent to an untreated piece of PA6

nanofibers are fully soluble. From a previously recorded calibration curve of the DR1 concentration (Supplementary Figure 15), we can determine the DR1 functionalization degree of the substrate (nmol/cm$^2$). All tests were performed on a 1-cm$^2$ piece of material in triplicate.

Initially, we investigated if the amount of adsorbed dye could affect the eventual dye loading after PDC by changing either the concentration of the dye in the dipping solution or the dipping time of the material (Supplementary Note 4). For this, DR1-allyl was dissolved in tetrahydrofuran (THF) with a concentration ranging from 10 to 100 mg/mL after which PA6 was dipped into these solutions for 1 to 60 min. The treated PA6 samples were subsequently dried in a vacuum oven for 30 min and plasma

treated for 10 min on both sides. The results show that the dye loading reaches a plateau at respectively 25 mg/mL dipping concentration and 1 min dipping time, allowing for a relatively high dye loading while limiting both time and spent resources during the first step of the PDC procedure. This behavior is probably related to the large spatial stability of the PA6 base material showing little swelling in THF, resulting in fast saturation of the surface with dye. This also means that although

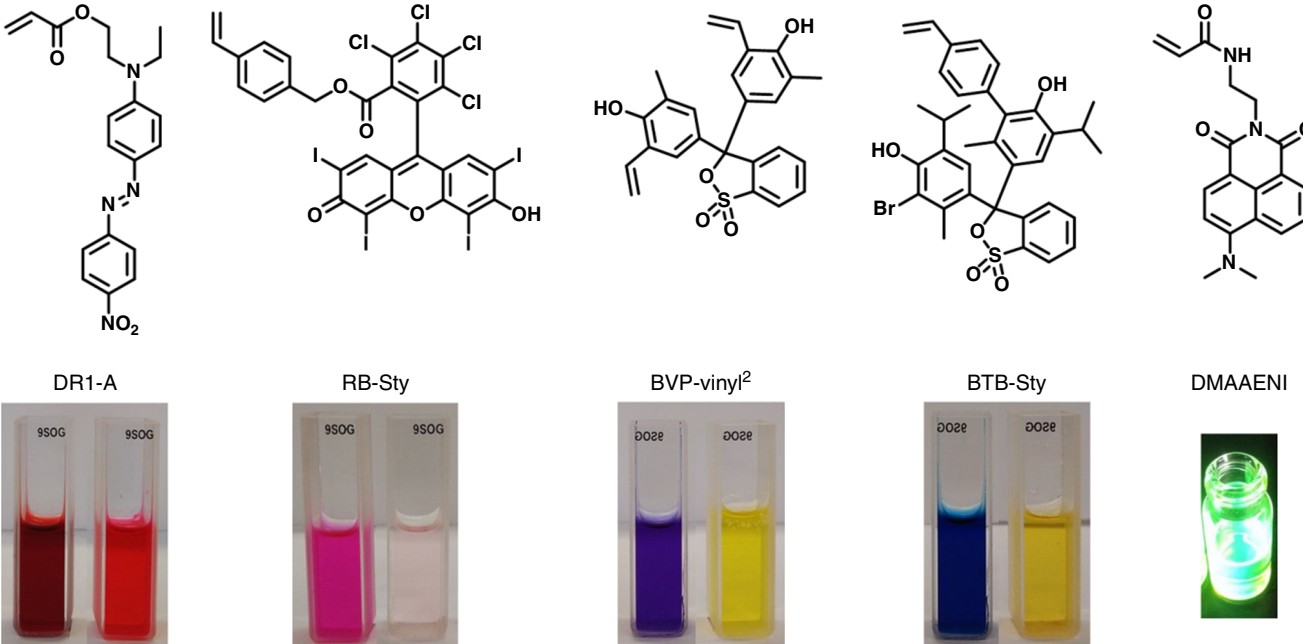

**Fig. 5** Overview of the structure, abbreviations, and visual properties of the used dyes. DR1-A, RB-Sty, BCP-vinyl[2], and BTB-Sty show halochromic properties from acidic (left) to basic (right) while DMAAENI shows strong fluorescence in chloroform

the dye loading is largely unaffected by concentration and dipping time, there could be a material specificity to the optimum conditions.

Perhaps the most important parameter to investigate is the effect of the plasma time as this controls the balance between dye immobilization and material/dye degradation. We investigated this by using the more easily polymerizable DR1-acrylate (DR1-A) dissolved in THF with PA6 and maintaining the previously optimized 25 mg/mL dipping concentration and 1 min dipping time. The evolution of the dye loading in function of the plasma treatment time on a single side (Fig. 3a) resembles this balance with an increase in dye immobilization up to a maximum around 1 min after which the degradation of the dye molecules and material becomes more important resulting in a steady decrease of dye loading. We can even visually notice this trend with the strongest orange-red color of the material after soxhlet extraction for the 1 min sample, which fades and becomes more brown with increasing plasma treatment time, indicative of dye degradation (Fig. 3b). The optimized PDC procedure for the immobilization of DR1-A on PA6 nanofibers consists of the following parameters: 25 mg/mL dye concentration, 1 min dipping time, and 1 min plasma treatment on both sides of the material.

Using the optimized procedural parameters, the importance of the presence and identity of the radical-sensitive functionality of the dye was investigated. First of all, these experiments revealed that the presence of the radical-sensitive group is of vital importance for the successful immobilization of the dye onto the material surface as without these groups no significant dye loading could be determined visually or spectroscopically (Fig. 4). In this case, the energetic Ar-plasma most likely interacts with the aromatic structure of DR1 or the dye in general, leading to fragmentation and loss of optical properties. The identity of the polymerizable group also influences the dye loading (Fig. 4a), which can possibly be related to the different radical stability for each group. Additionally, the relatively high amount of dye loading using DR1-A (50% increase compared to DR1-methacrylate (DR1-MA)) points to a strong correlation between the plasma time and the identity of the polymerizable group, indicating that future use of the PDC method will require the

optimization of the plasma treatment time for each radical sensitive moiety to obtain maximum immobilization efficiency. We have also evaluated an alternative method for increasing the dye loading through the addition of a radical cross-linker like pentaerythritol tetraacrylate (PETA). A preliminary test of this method adding a 50 mg/mL concentration into the dipping solution containing DR1-A showed an increase in dye loading of 100% through an increased reaction of surface radicals with polymerizable groups. Unfortunately, this method also causes the cross-linking of the nanofibers and the significant alteration of the material properties including solubility.

Finally, we have identified the presence of the polymerizable group on the dye and the generation of the surface radicals through PSM as the two key-components of our PDC procedure by performing reference experiments excluding these components (Fig. 4b). It is only when both are present that the PA6 material retains a clear orange color after soxhlet extraction (Fig. 4b). Both of the two counter examples confirm the necessity of the polymerizable group (Fig. 4b) and the plasma treatment (Fig. 4b) as no dye immobilization is observed after soxhlet extraction without these key-components.

**Investigation of PDC applicability toward dyes and materials.** The two vital components of the PDC procedure that were identified, i.e., a polymerizable dye and the PSM, may allow a wide variety of possible material/dye combinations that could be coupled using this procedure. In order to highlight this wide applicability, we selected high-end, in house produced materials as well as commercial materials and readily available everyday items for PDC treatment. Highly adsorbent materials including PA6 and cellulose could be used with the described procedure while more inert materials like PE, PP, and PTFE require an additional plasma treatment step prior to dip-coating in order to ensure the adsorption and homogenous distribution of the dye on the material surface.

Besides variation in the material, this procedure could also allow the introduction of an additional functionality to the material through careful selection of the dye. The previously used

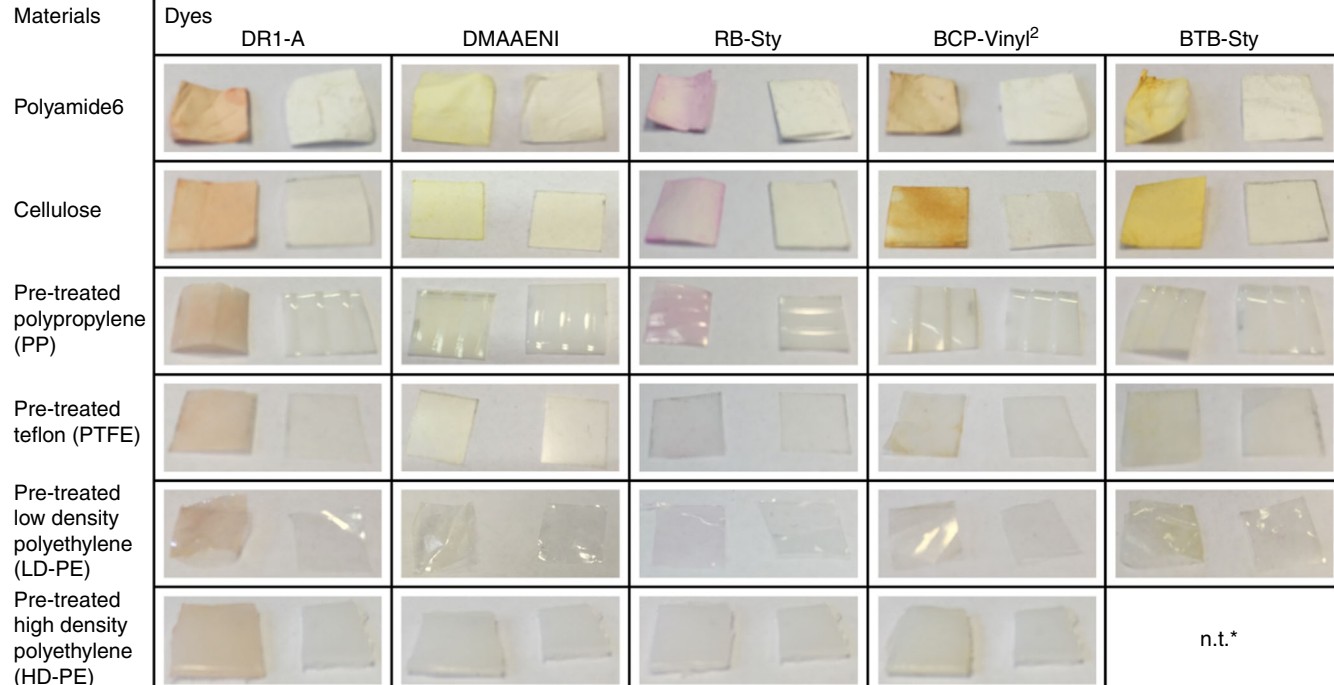

**Fig. 6** Overview scheme of the treated materials and applied dyes using the optimized PDC procedure. Each picture of the treated material after purification is taken compared to an untreated reference sample of the used material; * not treated due to insufficient amount of dye

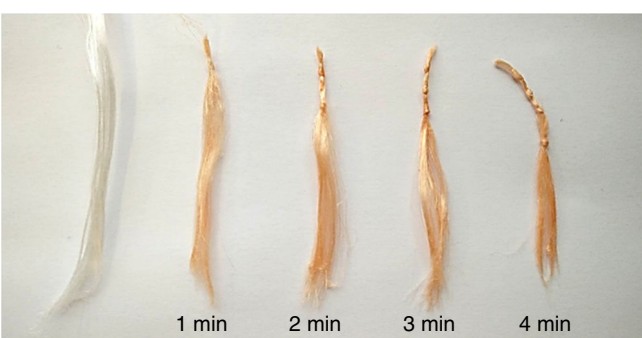

**Fig. 7** Optimization of the plasma treatment time for UHMW-PE fibrous material with DR1-A. Each picture of the treated fibrous material after purification is taken compared to an untreated reference sample of the used material

DR1 is known for its halochromic properties showing a color change from deep red to brightly pink at very acidic pH values[65]. These properties can be transferred onto the material by immobilizing these types of dye on its surface. In order to expand on this halochromic effect and highlight the wide applicability of the PDC procedure also in terms of the dye, we have selected a wide range of multiple complex halochromic and fluorescent dyes which were modified or synthesized to introduce radical sensitive groups (Fig. 5). Rose Bengal (RB) is characterized by a very bright pink color and halochromic transition from pink to orange at low pH values[66–68]. Another important family of pH-responsive dyes are based on the sulfonphthaleine structure, which are known for their good water solubility and tunable pKa through substituent selection, allowing the fine tuning of the pKa between 4.8 and 9.1[69]. In order to maintain the halochromic properties originating from the acidic phenol group, bromocresol purple (BCP) and bromothymol blue (BTB) were selected to be functionalized with styryl moieties through Stille and Suzuki-Miyaura coupling procedures onto their aryl-bromide

substituents, respectively. A final dye that was synthesized especially for this project is the solvatochromic fluorescent dye 4-$N$, $N'$-dimethylamino-1,8-acrylamidoethylnaphthilimide (DMAAENI) based on a reported synthesis of a similar dye by Inal et al.[70]. While DR1-A and DMAAENI are well soluble in THF at room temperature, BCP-vinyl[2], BTB-Sty, and RB-Sty required the use of ethanol in both the dip-coating step and the soxhlet extraction. The full experimental description of these syntheses can be found in the Supplementary Note 3.

The matrix shown in Fig. 6 highlights the wide applicability of the PDC procedure while also suggesting some more material and dye specificity than originally anticipated. Although all material and dye combinations are colored after soxhlet extraction, the intensity of the treated material varies significantly (Supplementary Note 5). The optimized PDC procedure seems to be most effective for the highly adsorptive materials cellulose and PA6, showing a relatively strong, homogenous color for all dyes. The plasma pre-treatment of the other materials not only increased the color intensity, but it also enabled the homogeneous distribution of the dye on the material surface. Especially for the very inert materials like untreated PP and PTFE, the dye solution would just bead off the surface and collect on the edges without any significant dye adsorption. Finally, when comparing the dyes it is obvious that the best results are obtained with DR1-A, which is expected as the PDC procedure was optimized with it. In general, the PDC procedure does allow the immobilization of a large variety of dyes with various radical sensitive groups on a wide range of materials. It is, however, apparent that the optimal PDC process parameters are both material and dye specific and that ultimately the dye loading for the other combinations besides PA6/DR1-A could be increased through further procedural optimization as demonstrated for LD-PE/RB-sty (Supplementary Note 6).

A major potential contributing factor to the difference in dye loading between the different materials is the surface area, especially when comparing fibrous materials like PA6 and cellulose filter paper with non-fibrous materials like PE and PTFE. To investigate the effect of surface area on the PDC process

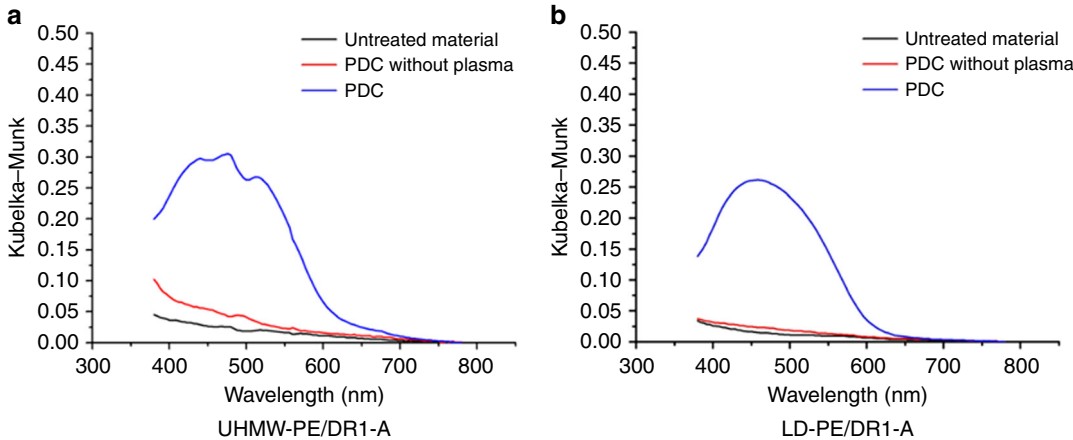

**Fig. 8** Reflective UV-VIS spectroscopy on UHMW-PE and LD-PE. **a** The Kubelka–Munk, normalized at max. reflection, for non-treated (untreated material), PDC without the plasma treatment step (PDC without plasma), and a PDC-treated (PDC) sample of fibrous UHMW-PE/DR1-A, performed under following PDC conditions: plasma time: 1 min, dipping concentration: 25 mg/mL, and dipping time: 1 min. **b** The Kubelka-Munk, normalized at max. reflection, are given for non-treated (untreated material), PDC without the plasma treatment step (PDC without plasma), and a PDC-treated (PDC) sample of non-fibrous LD-PE/DR1-A, performed under following PDC conditions: plasma time: 1 min, dipping concentration: 25 mg/mL, and dipping time: 1 min

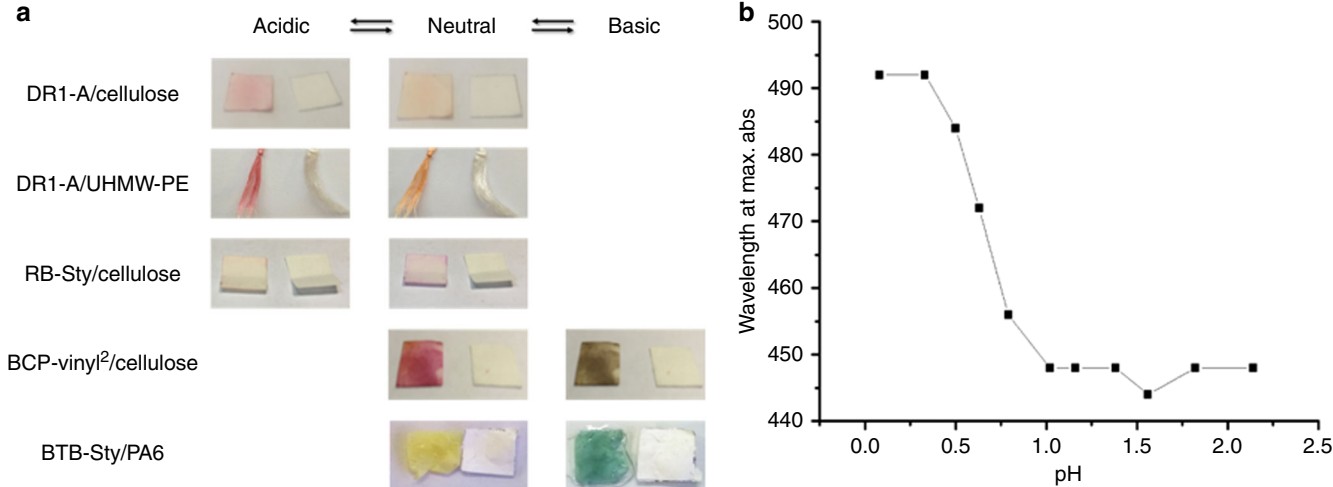

**Fig. 9** The halochromic behavior of plasma dye coated materials. **a** The halochromic behavior of these materials is induced by the application of HCl or NH$_3$ saturated vapor in the case of DR1-A, RB-Sty, and BCP-vincyl$^2$ or aqueous solutions in the case of BTB-sty. Each sample is photographed compared to an untreated reference sample. **b** Wavelength at max. absorbance in function of the pH dipping solution prior to reflective UV-VIS measurements of cellulose/DR1-A allowing the estimation of the pKa of DR1 around 0.56

by direct comparison of sheets and fibers of the same polymer material, ultra-high molecular weight PE fibers (UHMW-PE fibers; Dyneema®) were functionalized with DR1-A using the PDC procedure and the surface modification was compared to the flat LD-PE substrate (Fig. 7). The inert UHMW-PE fibers were PDC treated according to the same method as non-fibrous LD-PE, including the pre-treatment with plasma for homogenous dye adsorption during dip-coating.

Firstly, the inert fibrous UHMW-PE material is compatible with the PDC procedure as can be visually confirmed from Fig. 7. Secondly, the same trend showing higher dye loading with increased plasma time (Supplementary Note 7) can be observed for this material similarly to LD-PE (Supplementary Note 6). This observation indicates that the optimal PDC process parameters are both material and dye specific, but are less dependent on the surface area and shape of the substrate. Finally, a quantitative characterization of the dye loading was performed using reflective UV-VIS spectroscopy (Fig. 8). In both graphs, the increased

absorbance around 450–500 nm can be noticed proving the presence of the red dye molecule on the surface of the material after PDC treatment and soxhlet extraction. Although the trace of fibrous UHMW-PE is less smooth due to enhanced scattering resulting from the uneven fiber surface, the increased absorbance of 0.31 at the maximum absorption wavelength of DR-1 compared to 0.26 for LD-PE indicates a higher dye loading for the fibrous material under identical procedural PDC parameters (1 min plasma time). Further reflective UV-VIS measurements indicate that fibrous materials, like PA6, cellulose, and UHMW-PE materials, result in a higher dye loading because of a higher surface/bulk ratio compared to non-fibrous materials like Teflon and LD-PE (Supplementary Note 5). Finally, no distinction in intensity can be noticed between the PDC without plasma sample and the non-treated sample for non-fibrous and fibrous PE/DR1-A (Fig. 8) and PA6/DR1-A, Cellulose/DR1-A, Teflon/DR1-A (Supplementary Note 5) showing again the necessity of the plasma step in the PDC procedure.

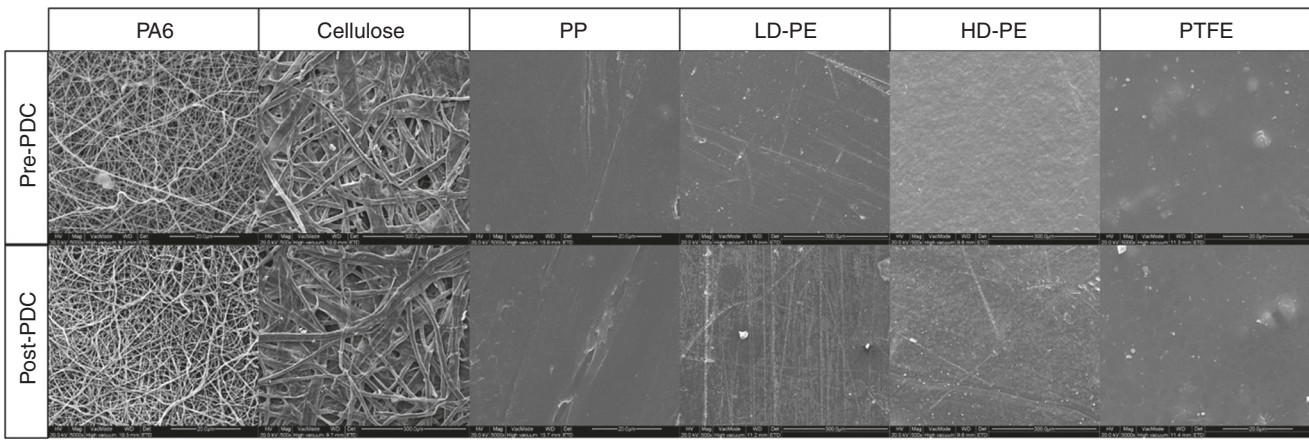

**Fig. 10** Overview of SEM images pre- and post-PDC treatment. All materials were coated with DR1-A using the PDC procedure and imaged after purification

The described post-fabrication modification approach firstly allows for the irreversible coloring of the material with no dye leaching, indicated by the remaining color on the material after extensive soxhlet extraction. We have also performed reflector UV-VIS spectroscopy and dye leaching tests to a water bath at different pH on DR1-modified PA6 samples (Supplementary Note 5 and Note 9, respectively) which further proves the irreversible dye immobilization. Secondly, the PDC procedure can be used to bestow an additional responsivity onto the material surface by careful design of the used polymerizable dye, allowing the use of the PDC procedure for the synthesis of various sensor materials. We have highlighted this possibility by fabricating a range of halochromic materials which could be used as colorimetric pH-responsive sensors. Examples of this color change in function of pH are illustrated in Fig. 9 where the first three examples are switched by the application of HCl or NH₃ saturated vapors due to the extreme pKa of the immobilized DR1 and RB dyes (pKa 0–1). The halochromic behavior for cellulose/DR1-A was analyzed via reflective UV-VIS by immersing the sample in different pH solutions varying from 0 to 2 prior to measurements (Fig. 9). The pKa of the PDC-treated sample is around 0.56 resulting from the titration curve, which is in line with the earlier found pKa of DR1 immobilized in fibrous materials as previously reported by our group[42]. We have also recorded the near instantaneous and reversible color change upon application of acidic or basic vapors in the movie added in the supplementary information showing the change of DR1-A immobilized on UHMW-PE using the PDC procedure (Supplementary Movie 1). The final sample of BTB-Sty/PA6 shows its less extreme pH-responsivity by adding distilled water (neutral) and an aqueous NaOH solution (pH 13; Fig. 9). All samples showed the desired quick and reversible homogenous color change, which makes these materials and the PDC procedure applicable for the fabrication of halochromic sensors.

To ensure the stability of the material during the PDC procedure, we have investigated the effect of the highly energetic plasma on the structural integrity and surface characteristics. Firstly, pictures were taken of the surface of all materials before and after PDC treatment using scanning electron microscopy (SEM), showing no significant influence of the plasma treatment on the surface (Fig. 10). The straight lines on both LD and HD-PE are mostly likely some tears or cuts from the manipulation of the material during the PDC procedure and SEM sample preparation as the PDC process should not lead to such artefacts.

Finally, the effect of the plasma treatment and dye immobilization on the material surface was further investigated by

comparing the hydrophilicity of treated and untreated samples using contact angle measurements. Values for untreated surfaces were compared to samples modified with the hydrophobic DR1 and the hydrophilic RB, immobilized using the PDC procedure. All tests were performed in triplicate by placing a drop of MilliQ water on each surface after which the contact angle was determined after 5 s, averaged over five drops per sample (Supplementary Figure 20). The results show that for most of the materials the influence of the PDC procedure on the surface properties is limited with neither a drastic change in the contact angles (Δ < 10°) nor a noticeable trend due to the incorporation of a specific hydrophobic (DR1) or hydrophilic dye (RB). The biggest difference is noticed for LD-PE showing a large decrease (30–40°) in contact angle for both PDC samples. This could be explained by the incorporation of oxygen species (alcohols or peroxides) upon opening of the plasma chamber to the air after PDC treatment. Alternatively, the use of organic solvents or increased temperature of the soxhlet solvent could have influenced the surface properties as it clearly affects the density and rigidity of the LD-PE. The results for PA6 and cellulose are not included in the graph since the water droplet was absorbed into the material within 5 s for all of the measured samples, showing no difference between treated or untreated samples.

The effect of the PDC treatment on the composition of the material surface was further investigated by using X-ray photoelectron spectroscopy (XPS) on various materials treated with RB-Sty. These results (Supplementary Figure 21) show no noticeable difference in O/C ratio of the untreated material and PDC-treated samples with varying plasma times (PA6 and LDPE) or optimized PDC conditions (cellulose, Teflon, and UHMWPE). This indicates that the chemical composition of the surface layer of the material is not fundamentally altered as a result of the PDC procedure, except for PTFE where the fluoride atoms have been cleaved off during the plasma pretreatment step. A more critical look at the molar ratios for PA6, UHMWPE fibers and LDPE (Supplementary Figure 21) show a slight increase in oxygen percentage which is probably due to exposure to the air after plasma treatment, as commonly observed for plasma treatment of polymers[61,62]. Surprisingly, no traces of the dye, indicated by the presence of I or Cl atoms (Fig. 5), are observed for the PDC-treated samples, indicating that the dye loading on the surface if insufficient for the XPS detector sensitivity (nmol/cm² range), despite that successful dye immobilization was confirmed visually and by reflective UV-vis spectroscopy. The full data obtained from the XPS measurement are shown in Supplementary Note 8.

## Discussion

In summary, this manuscript describes a novel post-fabrication modification approach for the coloring of polymeric materials based on the radical, covalent coupling of a pre-adsorbed polymerizable dye to the surface of a material through solvent-free PSM. This results in highly accessible dye functionalities on the material surface which is beneficial for sensing applications allowing fast and reversible responses to external stimuli. The PDC procedure shows successful immobilization for various dye–material combinations, allowing the creation of colored materials or sensors without the need of interfering with the fabrication method. This post-fabrication approach makes it possible to employ a single procedure for a wide variety of materials including fibrous materials like PA6 and cellulose, but also more inert, non-fibrous materials like PE and PTFE. The low amount of dye required and the possibility of reusing a dipping solutions results in a significant cost reduction allowing the use of more complex and expensive dyes and pigments. Other advantages include the absence of dye leaching and short treatment times. Finally, various surface-characterization techniques including SEM imaging, contact angle measurements, and XPS further indicate that the effect of the plasma treatment on the surface properties of the materials is less severe than we have anticipated, which is beneficial for future applications.

The economic potential of this PDC procedure lies in the possible usage of an atmospheric pressure plasma generator, allowing a continuous PDC process that could be directly linked to an existing industrial scale production line. Secondly, further optimization of the dip-coating step could include the use of other dry-deposition techniques like airbrushing to provide full surface coverage or to apply a specific design. Thirdly, the solvent-free modification step of the PDC procedure also limits its environmental impact in comparison to conventional dyeing methods. Finally, the optimization of other PDC steps could reduce the environmental impact even further by for example the use of green solvents like water and ethanol in the dip-coating and washing step, but also by downscaling the waste streams through reusing any used solvents.

## Methods

**Materials**. The PA6 support was fabricated in-house using electrospinning from acid solution according to a previously published protocol[42], cellulose-based qualitative filter paper purchased from VWR (Medium filtration rate), PTFE was obtained from Holders Technology, samples of high-density PE were cut out of a 2.5-L solvent container, low-density PE was cut out of small sealant bag, PP was cut out of a plastic cup, and finally the UHMW-PE fibers, also known as Dyneema® fibers, were obtained from DSM. Ethanol (absolute) and tetrahydrofuran (≥99.9%) used for the dipping solution and soxhlet extraction were bought from Fischer Scientific and Sigma-Aldrich, respectively, and used as received. The used dyes or starting compounds were used as received with DR1 (95%), BTB (95%), and BCP (90%) bought from Sigma-Aldrich, both Rose Bengal (acid red 94, 95%) and 4-bromo-1,8-naphthalic anhydride (95%) were bought from TCI Europe. Further synthetic details on the modification of these dyes can be found in the supplementary information (Supplementary Note 3). More information on materials can be found in Supplementary Note 1.

**Instrumentation**. The instrument used for the creation of the plasma was a cylindrical dielectrical plasma discharge generator (Femot Model, Version 3, Diener Electronic, Germany). Argon was used as discharge gas at a pressure of 0.8 mBar and was activated by a rf-generator (100 W). The dye loading during procedural optimization was determined by UV-VIS absorption spectroscopy of the PA6 sample after soxhlet extraction dissolved in formic acid/acetic acid (FA/AA) 1/1. For this, a calibration curve was determined using DR1 in the same acid mixture (Supplementary Note 4). DR1 is used as standard because the alcohol functionality used for modification of DR1 is isolated from the aromatic system and the effects of the modification and immobilization on the absorption spectrum are minimal. The contact angle measurements were performed on a SCA 20 Instrument (Dataphysics), equipped with a light source and high-speed video system with CCD cameras. To determine the static contact angles of the polymer surfaces, the sessile drop method was used. The static contact angle was determined 5 s after the first contact of the water droplet with the surface, using the circle fitting of the imaging software SCA20 (version 2.1.5). More information can be found in Supplementary Note 2.

**PDC procedure**. The optimized procedure for the PDC process of DR1-A on PA6 is as follows: a 1-cm² piece of PA6 is dipped into a 25 mg/mL solution of DR1-A in THF for 1 min. The material is carefully removed while residual THF is removed from the sides of the material using filter paper. The material is firstly dried to the air to ensure a homogenous dispersion of the dye after which the material is dried completely in a vacuum oven at 40 °C for 30 min. The material samples are then positioned in the plasma chamber on a glass surface and placed under 0.8 mBar of Ar pressure. After 1 min of plasma treatment, the chamber is opened to the air and the pieces are flipped over and treated again. The material is finally purified using soxhlet extraction using the same solvent as the dipping solution. For both DR1-A and the fluorescent dye DMAAENI, this means THF while rose bengal–styryl (RB-Sty), bromothymol blue–styryl (BTB-Sty), and bromocresol purple–divinyl (BCP-vinyl²) require the use of ethanol. Materials like cellulose and PA6 are sufficiently adsorbent to ensure a homogenous coverage of the polymer sample with the dye after dip-coating. PE, PP, and PTFE, however, were pretreated with plasma, under identical conditions as previously mentioned, before dip-coating to allow sufficient dye adsorption to the surface.

**Data availability**. The corresponding author acknowledge that the data supporting the findings of this study are available within the paper and its Supplementary Information files.

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

## Acknowledgements

The authors acknowledge the Agency for Innovation by Science and Technology of Flanders (IWT), the Research Foundation–Flanders (FWO), and Ghent University for financial support.

## Author contributions

The manuscript was written through contributions of all authors. G.V., L.D.S., and R.H. conceived and designed the procedure and experiments. L.D.S. and P.M. performed all of the syntheses and PDS experiments; E.S. performed the dye leaching experiments. K.L., I. S., K.D.C., E.V.D.W. and P.D. provided support and advice to synthesis, materials characterization, and plasma treatment, respectively. All authors have given approval to the final version of the manuscript. G.V. and L.D.S. contributed equally to the manuscript as joint first authors.

## Additional information

**Competing interests:** A patent application has been filed based on parts of this work for G.V. and R.H. (WO2017125462A1; Ghent University). The remaining authors declare no competing interests.

