## [Peer Review File(PDF 145 kb) · Nature Communications]

Reviewers' comments:

Reviewer #1 (Remarks to the Author):

Dear Editor, I have read thoroughly the manuscript of Vancoillie et al concerning the use of plasma activation for the grafting of dye molecules to different types of technologically relevant polymers. I think the approach use is interesting providing the process is directly up scalable to industrial fabrication. I think this is an example of study that fill the gap between the chemical approach and the physical methodologies. The authors proved the methodology is robust and present clear advantages with respect to traditional wet-chemistry methods. I think the publication of this paper can help to show there are many surface related methodologies that can be combined with chemical process and that is a field open for research.

There is a lack of surface chemical information of the original, treated and dye coated surfaces, this information should be included and discussed in the manuscript

My main concern with the manuscript content is the way the plasma-based processes are described. Plasma technology applied to material science and surface modification have shown a tremendous evolution in the last decades. The plasma deposition and treatment are very briefly and superficially described in the manuscript. At present, there are many monographies and revision papers describing the plasma treatment of polymeric surfaces and the plasma chemistry related, see for example Prog. Polym. Sci. 29(8) 815-885, 251 or Chem. Rev. 116, 3975-4005, 2016 as a recent example. On the other hand, the direct plasma polymerization of dye molecules to obtain insoluble and stable dye-containing films has been studied by several authors, see for example J. Phys. Chem. C 113 (1), 431-438, 2008 (insoluble films containing dyes for sensing), Adv. Mater. 23 (6), 761-765, 2011 (dye plasma polymers in photonic structures), ACS Applied Materials & Interfaces 9 (10), 8948-8959, 2017 (laser emission from dye containing plasma polymers). In my opinion the authors must rewrite several paragraphs and include those references and some others to show properly the potential of the plasma based methodologies an to show what are the importance of their result in this scientific context.

For the previous reason I recommend the publication but after MAJOR REVISION, I think the manuscript should be revised by referees after the mandatory modifications.

Reviewer #2 (Remarks to the Author):

This paper describes the use of plasma surface modification to immobilize dyes onto the surfaces of various materials. An aspect of the work is that the dye molecules have been functionalized with a “radical sensitive group” (i.e. a vinyl group) to improve grafting.

The technique is interesting, the paper is generally well-written and the conclusions are supported by the experiments.

The PA6 and cellulose are fibrous materials having significant surface area while the PP, PE, PTFA are monolithic plastics. It seems that not only is the chemical structure different, but the surface area is very different for these plastics. It would have been more appropriate to compare to fibrous versions of these plastics to the PA6 and cellulose fibers. In any event, the influence of the surface area should be discussed.

It would be good to have the chemical structures of the dyes and “radical sensitive groups” early in the article for reference.

Some specific comments:

Line 97: suggest changing “homolytical” to “homolytic”

Line 100: There is no reference to List et al.

Lines 131-133: This sentence could be clearer. Especially “that is applied being either” is awkward.

Line 145: “styrene” should be “styryl” (and elsewhere) since it is not styrene but a substituted styrene

Line 185: change “have investigate” to “investigated”

Line 190: what is degrading?

Line 205: It is not clear what “In the latter case” is referring to.

Line 236: Explain what “more resistant” means. More resistant to what?

Line 304: change “taking” to “taken”

Reviewer #3 (Remarks to the Author):

This manuscript reports a method for immobilizing dye molecules on substrates, using plasma to activate radical reactions. The methods, results, and interpretation are described clearly. The manuscript is well-written with only a few errors of language or typography.

While the work is potentially of practical applicability for higher-end products (given the cost of plasma treatment), I do not consider this manuscript to have the novelty I associate with a Nature family journal. There are no new fundamental scientific insights arising from this work; it's a methods paper rather than a study of mechanisms. I would consider this manuscript more suitable for a more applied polymer journal or a surface coating technology journal.

The novelty is somewhat limited also by the fact that plasma methods for grafting molecules including dyes via double bonds have been used for decades, and while to my knowledge the exact same work has not been reported previously, the work reported here is not that radically different from some ideas that are well-known in the surface coating community. The results, while nicely collected, are not really surprising and thus do not seem to me to be of the type that would spark a lot of follow-on work by many other labs.

In the introduction the authors refer to two current methods of adding dyes and discuss the disadvantage that the dye is only loosely attached to the material and thus susceptible to leaching. The authors then mention prevention of leaching, citing references 37 to 41 without giving details, and then discuss work only from within their group, going on to state that cost and time restrict that approach to high-end applications. This conclusion appears to ignore the fact that colour-fast textiles are being produced in large volumes at low cost. Their discussion of the state of the art seems somewhat biased to me, as there is extensive technology on dyeing textiles such as to obtain minimal leaching. I was wondering why the authors had not included discussion of methods in current use for the fabrication of materials that do not leach. If this omission was intended to emphasize the novelty of their work, then it would be misplaced.

The authors state on pages 7 and 8 that the dye is adsorbed on the material surface. Have they done experiments to exclude the possibility that some dye molecules diffuse into the material? Why

would the dye molecules prefer to adsorb on the surface in a multilayer fashion rather than diffusing into the PA6? What is the driving force for obtaining the presumed surface multilayers of dye?

Also, the interpretation is on crosslinking via radicals. But plasmas also possess strong UV emission. Can they exclude a role of UV irradiation?

Plasma Dye Coating: A Straightforward and Widely Applicable Procedure for Surface Dye Immobilization on Polymeric Materials

Reviewers' comments:

Reviewer #1 (Remarks to the Author):

Dear Editor, I have read thoroughly the manuscript of Vancoillie et al concerning the use of plasma activation for the grafting of dye molecules to different types of technologically relevant polymers. I think the approach use is interesting providing the process is directly up scalable to industrial fabrication. I think this is an example of study that fill the gap between the chemical approach and the physical methodologies. The authors proved the methodology is robust and present clear advantages with respect to traditional wet-chemistry methods. I think the publication of this paper can help to show there are many surface related methodologies that can be combined with chemical process and that is a field open for research.

There is a lack of surface chemical information of the original, treated and dye coated surfaces, this information should be included and discussed in the manuscript

Answer: We thank the reviewer for support of our work and for the constructive critical comment that too few surface characterisation techniques were used in the original manuscript. To address this, we have performed XPS and reflective UV-VIS measurements on PA-6, PTFE, cellulose, LD-PE and UHMWPE fibres to compare blank and control samples as well as PDC-treated samples using both DR1-A and RB-sty to identify and estimate the loading of the dye on the surface of the material. The XPS measurements have further proven that our PDC procedure only affects the surface without altering the bulk chemical properties of the polymeric materials. These results have been included in the revised manuscript (line 434-447) and the raw data has been included in the ESI (section S7).

My main concern with the manuscript content is the way the plasma-based processes are described. Plasma technology applied to material science and surface modification have shown a tremendous evolution in the last decades. The plasma deposition and treatment are very briefly and superficially described in the manuscript. At present, there are many monographies and revision papers describing the plasma treatment of polymeric surfaces and the plasma chemistry related, see for example Prog. Polym. Sci. 29(8) 815-885, 251 or Chem. Rev. 116, 3975-4005, 2016 as a recent example. On the other hand, the direct plasma polymerization of dye molecules to obtain insoluble and stable dye-containing films has been studied by several authors, see for example J. Phys. Chem. C 113 (1), 431-438, 2008 (insoluble films containing dyes for sensing), Adv. Mater. 23 (6), 761-765, 2011 (dye plasma polymers in photonic structures), ACS Applied Materials & Interfaces 9 (10), 8948-8959, 2017 (laser emission from dye containing plasma polymers). In my opinion the authors must rewrite several paragraphs and include those references and some others to show properly the potential of the plasma based methodologies and to show what are the importance of their result in this scientific context.

Answer: We agree that the original manuscript contained a rather brief introduction to plasma treatment of polymeric surfaces and related chemistry. However, we deliberately focussed the introduction on key aspects of this interesting and increasingly important field of material chemistry in order to publish a research article fit for a multi-disciplinary journal like

nature communications that appeals to both the experienced scientist in the area as well as to introduce the concept to the interested newcomer in the field of plasma surface modification.

We do however thank the reviewer for pointing out that the original introduction may have been too brief. The revised manuscript contains an extensively expanded introduction to explain plasma related processes more deeply and to introduce additional plasma based methodologies including plasma polymerization and plasma deposition. All suggested reference have been included.

For the previous reason I recommend the publication but after MAJOR REVISION, I think the manuscript should be revised by referees after the mandatory modifications.

Reviewer #2 (Remarks to the Author):

This paper describes the use of plasma surface modification to immobilize dyes onto the surfaces of various materials. An aspect of the work is that the dye molecules have been functionalized with a “radical sensitive group” (i.e. a vinyl group) to improve grafting. The technique is interesting, the paper is generally well-written and the conclusions are supported by the experiments.

Answer: We are grateful to the reviewer for the support of our work.

The PA6 and cellulose are fibrous materials having significant surface area while the PP, PE, PTFA are monolithic plastics. It seems that not only is the chemical structure different, but the surface area is very different for these plastics. It would have been more appropriate to compare to fibrous versions of these plastics to the PA6 and cellulose fibers. In any event, the influence of the surface area should be discussed.

Answer: We thank the reviewer for this excellent suggestion that we had not taken into account in the original manuscript. Indeed, the surface area per cm² of each material will play a major role in the eventual dye loading as the PDC allows covalent dye-coupling to the surface.

In order to introduce this concept in the revised manuscript, we have included the comparison between fibrous and non-fibrous PE materials and compared the dye-loading of RB-Sty after the PDC treatment using a visual comparison and reflective UV-VIS spectroscopy. The results confirm the material-dye specificity of the PDC procedural parameters as suggested in the original manuscript. Furthermore, significant higher dye loading is obtained on the PE fibers demonstrating a positive correlation between the dye loading and the surface area/bulk volume ratio. As such, the overall lower loading of non-fibrous PE, PP and PTFE compared to the fibrous PA6 and cellulose as reported in the original manuscript may be simply correlated to the surface area. The most important findings have been included in the revised manuscript and the effect of surface area on dye-loading is discussed (line 340-377); the raw data has been included in the ESI (Chapter S4 and Chapter S6).

It would be good to have the chemical structures of the dyes and “radical sensitive groups” early in the article for reference.

Answer: We agree that clarifying the concept of “radical sensitive groups” and the chemical structure of the used dyes early on in the manuscript will indeed help the reader to understand the importance of this key aspect of the PDC-procedure better. Therefore, the structure of the DR-1 dye with the different investigated ‘radical sensitive groups’ has been included early in the R&D part (line 196-204) while the other dyes are only shown when used to further broaden the scope of the PDC concept.

Some specific comments:

Line 97: suggest changing “homolytical” to “homolytic”

Line 100: There is no reference to List et al.

Lines 131-133: This sentence could be clearer. Especially “that is applied being either” is awkward.

Line 145: “styrene” should be “styryl” (and elsewhere) since it is not styrene but a substituted styrene

Line 185: change “have investigate” to “investigated”

Line 190: what is degrading?

Line 205: It is not clear what “In the latter case” is referring to.

Line 236: Explain what “more resistant” means. More resistant to what?

Line 304: change “taking” to “taken”

Answer: We appreciate the meticulous proof-reading of our manuscript and all these specific comments were addressed in the manuscript.

Reviewer #3 (Remarks to the Author):

This manuscript reports a method for immobilizing dye molecules on substrates, using plasma to activate radical reactions. The methods, results, and interpretation are described clearly. The manuscript is well-written with only a few errors of language or typography.

While the work is potentially of practical applicability for higher-end products (given the cost of plasma treatment), I do not consider this manuscript to have the novelty I associate with a Nature family journal. There are no new fundamental scientific insights arising from this work; it’s a methods paper rather than a study of mechanisms. I would consider this manuscript more suitable for a more applied polymer journal or a surface coating technology journal. The novelty is somewhat limited also by the fact that plasma methods for grafting molecules including dyes via double bonds have been used for decades, and while to my knowledge the exact same work has not been reported previously, the work reported here is not that radically different from some ideas that are well-known in the surface coating community. The results, while nicely collected, are not really surprising and thus do not seem to me to be of the type that would spark a lot of follow-on work by many other labs.

Answer: We acknowledge that surface grafting of monomers and polymers is known via plasma treatment, which we have described in the introduction chapter of the original manuscript and has been expanded in the revised version. Procedures like plasma-enhanced vapor deposition of dye molecules allows for the coloration of material surfaces by coating the material in a film in which the dye is imbedded through a complex plasma treatment procedure requiring specifically designed equipment. In this example procedure, the dye is introduced late in the plasma flow to prevent significant degradation of the dye and actually covers the material in a thin, colored film with altered surface properties rather than binding the dye to the surface. We are not aware of any report on a dye-plasma coating that directly binds the dyes covalently to the material surface in a single plasma treatment step as this is limited by the stability of the dyes under these conditions.

The important fundamental knowledge and insights resulting from this work is that the introduction of a radical sensitive group in the form of a double bond in the dye acts as an antenna for the reactive species generated by plasma leaving the majority of the dyes intact, at least at low plasma treatment times. The novel insight is that even though plasma contains highly reactive species, they mostly react with the antenna and do not significantly degrade the dyes allowing for a simple and direct method for covalently immobilizing the dye on the material surface.

From the critical remarks it appears that this antenna effect was not clear enough from the original manuscript and we have more explicitly discussed it the revised manuscript (line 174-177).

In the introduction the authors refer to two current methods of adding dyes and discuss the disadvantage that the dye is only loosely attached to the material and thus susceptible to leaching. The authors then mention prevention of leaching, citing references 37 to 41 without giving details, and then discuss work only from within their group, going on to state that cost and time restrict that approach to high-end applications. This conclusion appears to ignore the fact that colour-fast textiles are being produced in large volumes at low cost. Their discussion of the state of the art seems somewhat biased to me, as there is extensive technology on dyeing textiles such as to obtain minimal leaching. I was wondering why the authors had not included discussion of methods in current use for the fabrication of materials that do not leach. If this omission was intended to emphasize the novelty of their work, then it would be misplaced.

Answer: We thank the reviewer for pointing this out as it has never been our intention to bias the introduction and we would like to refer to the second comment of the first reviewer regarding our brief introduction that was intended to serve the broad interest of the readership of Nature Communications.

We are of course aware of the traditional textile coloration technologies that allow for low-cost and colour-fast textiles, such as the use of reactive dyes or addition of fixation agents, which has been further clarified in the revised manuscript (line 68-71). However these methods are typically based on an optimal tuning of the dyeing process for each polymer type to ensure mostly physical binding (more specifically woven fibers and not flat sheets) and are, moreover, not applicable to inert polymer surfaces such as PE, PP and Teflon. In contrast, our newly developed method can be applied on any polymer substrate, independent of shape and composition.

The authors state on pages 7 and 8 that the dye is adsorbed on the material surface. Have they done experiments to exclude the possibility that some dye molecules diffuse into the material? Why would the dye molecules prefer to adsorb on the surface in a multilayer fashion rather than diffusing into the PA6? What is the driving force for obtaining the presumed surface multilayers of dye?

Answer: Even though we have not experimentally verified that the dyes only retain at the surface, the short dipping time in combination with using a solvent that does not dissolve the polymer materials is assumed to significantly limit dye diffusion into the polymer materials. Furthermore, our control experiment of a PA sample with adsorbed dyes without plasma treatment (Figure 4c) clearly shows that there is not dye fixation. This has been further clarified in the manuscript (line 208-211).

Also, the interpretation is on crosslinking via radicals. But plasmas also possess strong UV emission. Can they exclude a role of UV irradiation?

Answer: As the introduction of radical sensitive groups leads to significant dye coupling it is most likely a radical process. Furthermore, the use of strong UV-irradiation would most likely degrade the dyes as they preferentially absorb this making it unlikely that UV is the main contributor for the observed dye immobilization.

REVIEWERS' COMMENTS:

Reviewer #1 (Remarks to the Author):

Dear Editor, I think the authors have improved the overall quality of the manuscript so I would like to recommend its publication. Some minor aspects as the scalability of the technique to industrial fabrication and the potential of the technique to reduce the environmental impacts of current fabrication process should be emphasized.

Reviewer #2 (Remarks to the Author):

The authors have satisfactorily addressed my comments and the article is significantly improved.

Reviewer #3 (Remarks to the Author):

Editorial Note: this Reviewer provided comments to the Editor only

Plasma Dye Coating: A Straightforward and Widely Applicable Procedure for Surface Dye Immobilization on Polymeric Materials

Reviewers' comments:

Reviewer #1 (Remarks to the Author):

Dear Editor, I think the authors have improved the overall quality of the manuscript so I would like to recommend its publication. Some minor aspect as the scalability of the technique to industrial fabrication and the potential of the technique to reduce the environmental impacts of current fabrication process should be emphasized.

We thank the reviewer for his kind words. The comment of the reviewer has been incorporated by including an extra paragraph in the discussion part which emphasizes more the economic potential and the outlook of the PDC technique. (line 435-443)

Reviewer #2 (Remarks to the Author):

The authors have satisfactorily addressed my comments and the article is significant improved.

We thank the reviewer for support of our work and for the initial constructive critical comments.

Reviewer #3 (Remarks to the Author):

Editorial Note: this Reviewer provided comments to the Editor only.